



# The Modern Ocean Sediment Archive and Inventory of Carbon (MOSAIC): version 2.0

Sarah Paradis[1], Kai Nakajima[1], Tessa S. Van der Voort[2], Hannah Gies[1], Aline Wildberger[1], Thomas Blattmann[1], Lisa Bröder[1], Tim Eglinton[1]

[1]Department of Earth Sciences, Geological Institute, ETH Zürich, Sonneggstrasse 5, 8092 Zürich, Switzerland
[2]Nutrient Management Institute, 6709PA Wageningen, Netherlands

*Correspondence to*: Sarah Paradis (sparadis@ethz.ch)

**Abstract.** Marine sediments play a crucial role in the global carbon cycle by acting as the ultimate sink of both terrestrial and marine organic carbon. To understand the spatiotemporal variability in the content, sources and dynamics of organic carbon

in marine sediments, a curated and harmonized database of organic carbon and associated parameters is needed, which has prompted the development of the Modern Ocean Sediment Archive and Inventory of Carbon (MOSAIC) database. We present here MOSAIC version 2.0, which has expanded the spatiotemporal coverage of the original database by ~400 %, and now holds data from more than 17000 individual sediment cores from different continental margins on a global scale. Additional variables have also been incorporated into MOSAIC v.2.0 that are crucial to interpret the distribution of the quantity, origin

and age of organic carbon in marine sediments globally. Sedimentological parameters (e.g., grain size fractions, mineral surface area) help understand the effect of hydrodynamic sorting and mineral protection in the distribution of organic carbon, while molecular biomarker signatures (e.g., lignins phenols, fatty acids, alkanes) can constrain the origin of organic matter with a greater level of specificity. MOSAIC v.2.0 also stores data from analyses performed on specific sediment and molecular fractions, which provide further insight of the processes that affect the degradation and ageing of organic carbon in marine

sediments. While data included within MOSAIC is not exhaustive, it is continuously expanding and version control will allow users to benefit from updated versions while ensuring reproducibility of their findings.

## 1 Introduction

Marine sediments are the ultimate sink of particulate organic carbon (OC), and play a fundamental role in the global carbon cycle. Understanding the functioning of the carbon cycle requires investigations of the distribution, composition and dynamics

of OC in marine sediments on different spatial and temporal scales. However, given resource and time limitations, studies prioritize either their spatial breadth and/or the factors and parameters measured (e.g., Bao et al., 2016; Mollenhauer et al., 2004; Smeaton et al., 2021). This leads to dispersed and unstandardized datasets that are often specific to individual research questions and/or laboratories, hindering broader assessments and upscaling of findings.

Proper harmonization of marine sedimentary data is especially important given the logistical challenges and costs of retrieving

samples at sea. Compiling a globally distributed dataset of OC and its geochemical composition is crucial to understand large-





scale patterns that affect its distribution, and has therefore been an ongoing effort for the past decades. The first global maps of the distribution of organic matter in surface sediments was presented by Premuzic et al. (1982) and Romankevich (1984), but these essentially highlighted patterns in the global distribution of OC, without providing accurate estimates. With the advance of geostatistical techniques, a more precise distribution of surficial OC content was performed by Seiter et al. (2004)

using over 5500 datapoints. However, geostatistical techniques can only infer the OC content in areas where data is available, since they rely on neighbouring datapoints to perform a kriging interpolation (Oliver and Webster, 1990). The onset of spatial machine learning techniques in the field of geosciences has allowed the interpolation of OC contents in unsampled areas that present similar explanatory features (i.e., surface ocean primary productivity, oxygen concentrations, sedimentation rates, etc.). This was first undertaken by Lee et al. (2019) using a slightly expanded dataset of 5600 datapoints, and revisited by Atwood

et al. (2020) using 11500 datapoints. While all of these studies agree that higher OC contents are found on continental margins in comparison to the open ocean, these margins are highly complex and temporally heterogenous, which is why more efforts should be directed to compiling data in these areas. With the exception of Seiter et al. (2004), these studies do not report their raw data, which not only hinders assessment of the reproducibility of their findings, but does not adhere to the Findability, Accessibility, Interoperability and Reusability (FAIR) data principles (Wilkinson et al., 2016).

The availability of a harmonized database of OC in marine sediments is crucial not only to refine estimates of carbon stocks of maritime nations (Avelar et al., 2017; Luisetti et al., 2020; Smeaton et al., 2021) and in marine protected areas (Atwood et al., 2020). Compositional information of sedimentary organic matter, such as its isotopic ($^{13}$C and $^{14}$C) and elemental composition, can help define spatial patterns in the distribution of the source and age of OC in marine sediments (Galy et al., 2007; Kao et al., 2014; Van der Voort et al., 2018), as well as determine its reactivity (DeMaster et al., 2021). With this in

mind, compiling and harmonizing diverse types of data is fundamental to understand the spatial variations in the content, source and composition of OC in marine sediments. Hence, the Modern Ocean Sedimentary Archive and Inventory of Carbon (MOSAIC) was constructed, with data on the content (%OC), stable carbon isotope ($\delta^{13}$C), radiocarbon ($\Delta^{14}$C), elemental (C:N ratio) composition of OC, as well as biogenic silica and CaCO$_3$ contents of marine sediments in 4460 locations worldwide (van der Voort et al., 2021).

Since the publication of the initial MOSAIC database, new metadata reporting strategies have stressed on the importance of standardizing measurement techniques (Morrill et al., 2021). Indeed, different measurement techniques yield different values, that can be up to 25 % different depending on the method employed (Byers et al., 1978; Celia Magno et al., 2018; Hoogsteen et al., 2018; Schubert and Nielsen, 2000), which could jeopardize proper comparability between studies and laboratories (Wilkinson et al., 2016). To maximize the comparability of data, the metadata reported in MOSAIC was revised and updated.

Moreover, to further understand the role of hydrodynamic sorting and mineral protection in the distribution of OC (Ausín et al., 2021; Bao et al., 2019; Bruni et al., 2022; Hemingway et al., 2019), as well as to assess the dispersal of specific sources of terrigenous OC in marine sediments (Gordon and Goñi, 2004; Hou et al., 2020; Yu et al., 2021) and contrast the age of OC compound classes (Hou et al., 2021; Kusch et al., 2021), the MOSAIC database was expanded to incorporate sedimentological properties (e.g., grain size distribution, mineral-specific surface area, porosity) and biomarker concentrations (e.g., lignin



phenols, fatty acids, alkenones), as well as data from the analyses on specific sedimentary components (i.e., grain size or density fractions, specific organic compounds). Finally, the spatiotemporal coverage of the database has more than quadrupled since the publication of the first MOSAIC iteration (van der Voort et al., 2021).

These changes have prompted us to publish a new version of MOSAIC (v.2.0), with an updated metadata structure and automated ingestion pipeline (see section 2), additional variables (see section 3), and expanded spatiotemporal coverage (see

section 4).

## 2 MOSAIC v.2.0 design

With the purpose of expanding the database's breadth and utility, the content in MOSAIC was revised and expanded, the database schema was restructured, and the pipeline for the incorporation of new data was improved. For instance, it is common for marine studies to present previously published data to provide greater spatiotemporal contextualization of the new findings

(i.e. Bao et al., 2016; Goñi et al., 2006; Gordon and Goñi, 2003; Kao et al., 2014). However, if data is independently added to the database, it can lead to large amounts of duplicate data entries, which in turn can skew the global dataset. Indeed, a careful re-examination of the first MOSAIC database revealed that ~20 % of the OC data and ~25 % of the $^{14}$C data was duplicated. Similarly, the same sediment samples can be analysed for different parameters (e.g., OC content, sediment grain size, specific biomarkers) and the results of these different analyses are often presented in separate studies (e.g., Bröder et al., 2016; Vonk

et al., 2012). When ingesting the data separately, they would be registered as separate samples, and therefore comparison of relationships between these variables would be hindered. Finally, inherent limitations of the length and precision of certain data types led to the loss of data (i.e., when surpassing maximum varying characters or maximum integer length), whereas coordinates of certain samples were found to be incorrect.

To overcome these issues, the structure and ingestion pipeline of the database was amended such that, to the best of our

knowledge, data is properly georeferenced, data duplication and data loss is minimized, and data comparability is improved. Given the ever-expanding body of data that continues to be published, we acknowledge that the structure and pipeline of the database will require further tuning and revisions based on user feedback and our experience.

## 2.1 MOSAIC v.2.0 structure

MOSAIC is a normalized relational database that creates separate tables which are related to each other to avoid redundancies

and efficiently store data. For instance, instead of storing data related to the sampling location in every subsample of a sediment core, it is stored only once in a separate table. The relations within the MOSAIC database follow a hierarchical list of tables that can be grouped into article and author (source metadata), Geopoints (location and sampling metadata), and samples (analyses) (Fig. 1). For this new iteration, the database was migrated from MySQL to PostgreSQL, which holds more advanced and efficient Geographical Information System (GIS) functions through its PostGIS extension. To increase the data storage

efficiency, several many-to-many relationships were built, such as between articles and authors (Fig. 1). In addition to these





minor modifications, MOSAIC v.2.0 has additional tables to incorporate new variables (Fig. 1) (see section 3). To increase the utility of the database, a few changes were made in the reported metadata to overcome the conundrum of sharing unstandardized data (Borgman, 2012). Some of the issues addressed in this new database are outlined below.

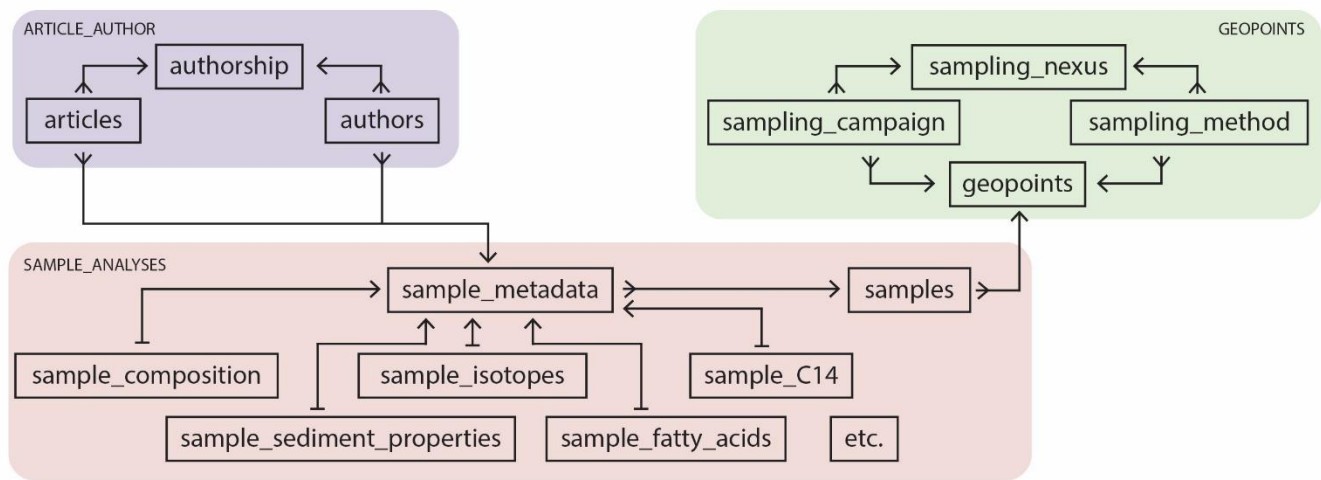

**Figure 1. Schematic representation of the structure of MOSAIC v.2.0, its tables and inter-relationships, omitting primary keys and foreign keys for simplicity.**

Despite relying on spatial data analysis, some studies in the field of geosciences still report their sample locations only through maps rather than providing their coordinates in tabular format (e.g., Guo et al., 2021; Hu et al., 2013; Pedersen et al., 1992; Zuo et al., 1991), hindering their addition to a harmonized database and contributing to the long-tail of lost data in marine

geoscience. With the improvement of map georeferencing tools (Hackeloeer et al., 2014), the locations of some of these sampled cores can be salvaged, resulting in an increase in the number of sediment cores included in MOSAIC v.2.0 by > 1000 (Fig. S1). However, the accuracy of these georeferenced datapoints may be less precise than those directly reported, so it is important to distinguish the source of the coordinates ("georeferenced_coordinates" column in Geopoints table). Similarly, studies seldomly specify the exact sampling date of their samples, hindering a proper analysis of the temporal variations in

sediment characteristics. The inconsistent reporting strategy of sampling date complicates the storage of this information in the database. One way to overcome this issue is separating the sampling date into year, month, and day, to allow users to add as much temporal detail as possible, which has been done in other geospatial databases (Ke et al., 2022). For a full list of metadata attributes available for Geopoints, see Table S1.

Following sample metadata-reporting strategies of analyses in other fields of geosciences (Morrill et al., 2021), the new version

of the database includes a hierarchical explanation of the measurement technique used to obtain the data: a first high-level category classifies the general method employed to obtain the data, while the second level allows a free-text entry of specific details used in the method (Table S2, Dataset S1). The category of the methods for each variable was established and discussed through extensive bibliographic research and the guidance of experts in the field. This categorical variable allows for quick comparison of data obtained using different methods, or to filter data based on the general method employed. For instance,



data of OC content can be obtained through an elemental analyser, mass loss through combustion, coulometry, titration, or manometric measurements, while information in the second category would allow a more detailed explanation of the sample pre-treatment, specific equipment model employed, or its instrumental settings (Fig. S2). In the case of grain size analysis, the most commonly used methods are sieving, particle settling time using Stokes' Law, and laser diffraction, which can provide variable results depending on the characteristics of the sediment (Celia Magno et al., 2018), while the user could specify the

sample pre-treatment (i.e., combustion, wet oxidation) in the method details (Fig S2). In the case of biomarker data, such as for alkanes and fatty acids, we encourage the user to add the measured homologues/chain lengths in the method detail in order to facilitate comparability between studies. Since this database is a collaborative effort, if an important category of the analytical method is not included in MOSAIC, we urge researchers to contact us so that it may be incorporated into future versions.

MOSAIC v.2.0 allows the specification of the material or fraction analysed for each measurement conducted (e.g., bulk sediment, OC, grain size, density, specific compound class or fraction) as "material_analyzed" (Table S3). This enables efficient storage and query of the analyses performed on the same sample but on different fractions, leading to a quick comparison of the OC content and isotopic composition in different grain size or sediment density fractions to assess, for instance, the effect of hydrodynamic processes (Ausín et al., 2021; Bao et al., 2019; Bruni et al., 2022), or the radiocarbon

contents of different compounds to understand the origin, reactivity and/or transit time of OC through the system (Eglinton et al., 2021; Kusch et al., 2021). Finally, the database also indicates if a value is a replicate analysis of the same sample, rather than providing a different identifier.

MOSAIC v.2.0 also improves the link of the reference (e.g., DOI) for each analysis. In the previous database structure, a sample was associated with one reference. Consequently, if the same sample was presented in two different studies with

complementary analyses (e.g., sedimentological and biomarker properties), they would be assigned different sample identifiers, hindering assessment of relationships between these analyses. To overcome this and incorporate the additional metadata (methods, material analysed), a new sample metadata table was devised that allowed pairing different analyses together while retaining their respective reference, measurement method, and material analysed (Table S2). In this metadata table, each row represents a specific measurement conducted on a specific sample, material (fraction) analysed, method,

reference, and replicate, if applicable. This structure allows the same sample to be assigned different references based on the type of analysis conducted, method employed, and/or material analysed. Finally, an additional column indicates whether the value was provided by the user or was calculated through harmonization techniques (see section 2.2.3).

## 2.2 MOSAIC v.2.0 data ingestion pipeline

The pipeline for the incorporation of data into MOSAIC v.2.0 follows a similar format as in the previous version, with a first

step of data ingestion, followed by a quality check, then its population to the database, and finally a user-friendly website where data can be visualized and extracted. In this section, we outline the changes applied to each step of the pipeline, and the



reasons that motivated them. The data ingestion template and scripts that automate the quality check and database population can be found in the GitHub repository (https://sarah-paradis.github.io/MOSAIC/).

### 2.2.1 Data ingestion

As with the previous iteration of MOSAIC, an Excel template workbook is provided, with separate sheets based on the type of data to be submitted (article where data is stored, information about the sampling location, and analyses conducted). With the expansion of the number of variables included in MOSAIC v.2.0, the previous spreadsheet file needed to be modified to avoid an excessive number of columns. Instead, the variables are classified based on their tables (Dataset S2) and a drop-down menu allows users to select which variables they want to provide, allowing users to accommodate the template according to

their dataset. Once the user has chosen a variable to be ingested, another drop-down menu appears with the list of general methods that the database accepts, as well as the material/fraction analysed (see section 2.1). This data ingestion workbook can be downloaded either from its GitHub repository (https://sarah-paradis.github.io/MOSAIC/) or from the MOSAIC website (mosaic.ethz.ch). The GitHub repository also provides a tutorial on how to fill in the template, along with an example workbook (https://sarah-paradis.github.io/MOSAIC/excel_template_tutorial.html).

### 165 2.2.2 Data quality check

The previous quality check structure simply determined if the data provided of each variable was within a specified (i.e., plausible) range. In this new ingestion pipeline, the Python script was expanded to raise an error if the data is not in a specified format or is not within a specified range. If specified, the algorithm also compares the data with data stored in other columns and raises an error if the criteria are not met. For instance, this comparison ensures that the error values are lower than the

variable value itself, or that the sum of certain variables equal a value (i.e., the sum of grain size fractions cannot be greater than 100 %), when appropriate. A warning message is raised if the data are not within a common range so that the curator can assess and, if necessary, correct the values. The full list of quality check parameters is provided in Dataset S1.

The script not only checks the values of the variables, but also inspects if all required fields (i.e., core name, sample name, exclusivity clause, etc.) are provided. In the case of article information, the script automatically extracts corresponding

metadata stored in Cross-Ref using the article's Digital Object Identifier (DOI), if provided. To prevent errors in the geographical positioning, the algorithm checks if the cores are located in the ocean using the NOAA high resolution coastline GSHH v.2.3.7 product (Wessel and Smith, 1996), and adds complementary geospatial information such as its Sea (Flanders Marine Institute, 2018), Exclusive Economic Zone (EEZ) (Flanders Marine Institute, 2019), Longhurst province (Longhurst et al., 1995), and MARCAT code (Laruelle et al., 2013). If the water depth is not provided, the algorithm extracts data from

the GEBCO bathymetric database (GEBCO Compilation Group, 2022) and specifies this in the Geopoint metadata (Table S2).



### 2.2.3 Data population

To upload data in the previous iteration of MOSAIC, a Python script separated the input template based on the individual SQL tables, which then had to be manually uploaded to the database (van der Voort et al., 2021). However, this process is tedious and does not check if the data already exist in the database, potentially leading to duplicate entries. In MOSAIC v.2.0, a new

Python script automates the data ingestion while querying the database to prevent duplicate data. This process is explained in more detail below for each level of the database's hierarchy (Fig. 2).

The population workflow of the article and author sheet is summarized in Figure 2a. This population is best achieved when the DOI is provided since the script extracts the standardized metadata from Cross-Ref, a repository of research objects that use DOIs, or PANGAEA (Diepenbroek et al., 2002), the biggest earth science repository. This also allows automatic population

of all the co-authors of each study, without requiring the user to upload this information, which can be tedious if the manuscript has many co-authors. However, if the DOI is not provided, the script can still populate all the provided data. This data population is done for each row, and iterates through the sheet by first populating the information related to the manuscript (title, year of publication, journal, and DOI) and assigning it an identifier. The script then iterates through the authors and assigns an identifier to each author, and finally creates the authorship table that stores data of this many-to-many relationship

(Fig. 2a). Throughout this population, the script queries the database to ensure that the information that is being added is not already in it.

Population of the data associated to each location (Geopoint) is similarly managed (Fig. 2b), but requires overcoming an additional handicap. As mentioned previously, given the high costs and complicated logistics of oceanographic cruises, these sampling campaigns are often conducted to achieve several research goals. Hence, sediment cores are often collected to

conduct different analyses on the same samples, leading to the publication of data originating from the same sediment core in independent manuscripts (e.g., Palanques et al., 2022; Paradis et al., 2021). Unfortunately, the same sediment core may be published using different naming conventions (e.g., Goñi et al., 2006; Gordon and Goñi, 2003), which complicates assignment of the correct Geopoint identifier to the new sediment core. Unfortunately, this may not be circumvented by matching either the coordinates or the sampling date since studies report their coordinates in different units (decimal degrees, decimal-minutes,

or decimal-minutes-second), and with different precision, whereas sampling date is often not fully provided (e.g., only the year or month if often reported). For instance, the outcomes of EUROSTRATAFORM project were published in different studies, with different coordinate precisions and slight variations in the naming convention (Kiriakoulakis et al., 2011; Masson et al., 2010; de Stigter et al., 2007). To account for this, the new population algorithm first queries the database to check if the exact same core name and coordinates is already available. If not, rather than assuming that the core is not in the database, the

algorithm then queries if there are any nearby cores, within 0.6 arc minutes (~ 1 km at the equator), and if so, it prompts the curator to check whether any nearby cores are actually the same (Fig. 2e).



**Figure 2. Schematic diagrams of data population into MOSAIC v.2.0. Population workflow for a) article information, b) Geopoint locations, and c) sample analyses. d) Population workflow of tables to avoid duplicates and link different analyses to the same sample. Colors in the tables indicate similarities in the values between each row (Data in MOSAIC vs. Data to be added). e) Database query**





**for nearby cores. f) Sample analysis table grouping to populate each individual table. g) Example of harmonization workflow of radiocarbon analyses and sample composition.**

The matching of Geopoints allows different analyses presented in separate studies to be linked, enhancing the scientific richness of the database. A similar protocol is applied to the sample analysis data (Fig. 2c), but this is further complicated by
the complexity of corresponding metadata given that the structure of this new database allows the specification of the material that is analysed (i.e., bulk sediment, OC, grain size fractions, compound specific analyses), as well as the method employed. Hence, before populating the database, the data is first separated based on the material analysed and methods employed, to allow for efficient storage of this metadata (Fig. 2f, Table S2). To further enrich MOSAIC v.2.0, automatic calculations harmonize and expand the database during the ingestion (Fig. 2g; Table S4). For instance, this new version implements
calculations to harmonize radiocarbon data between fraction modern ($F^{14}C$), $\Delta^{14}C$ and radiocarbon age, as defined by Stuiver and Polach (1977), and as specified in the previous MOSAIC version (van der Voort et al., 2021). The sample metadata then stores whether it was calculated through this data harmonization step (calculated data) or whether it was provided in the publication (reported data). This harmonization is performed for each group of data, and then populates the SQL table. Since all the variables stored in a table are not always provided in a study, the data population workflow should allow complementary
analyses to be added to the same sample. To do so, the population algorithm goes through a series of steps, adding complementary analyses if these are missing in the database, or by prompting the curator to take action (replace or assign as replicate) if the data to be added differ from the data that are already in the database (Fig. 2c).

## 3 Additional variables in MOSAIC v.2.0

The number of variables stored in MOSAIC v.2.0 increased by ten-fold in comparison to the previous iteration (Fig. S3), which
only contained data of OC, total and organic nitrogen, $CaCO_3$, biogenic silica, and the isotopic composition of OC ($^{13}C$ and $^{14}C$) (van der Voort et al., 2021). These initial variables are crucial to understand variations in the geochemical signature due to degradation and ageing processes of OC (Fig. 3a) or its sources, since contrasting fractionation processes and radioactive decay lead to distinct isotopic signatures of OC depending on its source and history (Fig. 3b). As many more factors can also affect the distribution of OC in marine sediments (Bianchi et al., 2018; Blair and Aller, 2012), MOSAIC v.2.0 incorporates
additional variables, including sedimentological parameters (section 3.1) and specific biomarkers (section 3.2).



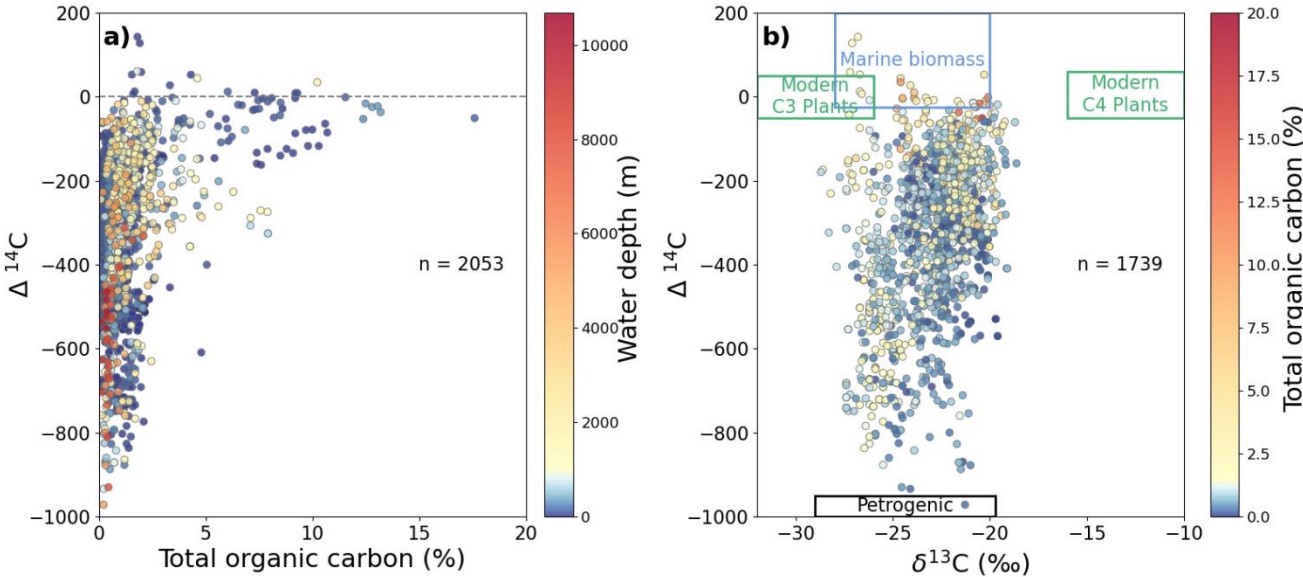

**Figure 3. a) Relationship between OC content and $\Delta^{14}$C based on water depth. Note that in shallow environments (< 200 m), there is a wide variability of OC content and age, whereas in deeper settings, OC and radiocarbon content converge to lower content and older ages. b) Scatter plot of the isotopic composition ($^{13}$C and $^{14}$C) of OC in surface sediment stored in MOSAIC v.2.0 and the isotopic signatures of the main end-members: marine biomass (Verwega et al., 2021), C3 and C4 plants (Bender, 1971; Farquhar et al., 1989), petrogenic organic carbon (Copard et al., 2022; Hilton et al., 2010; Walinsky et al., 2009).**

### 3.1 Sedimentological properties

One of the additions in MOSAIC v.2.0 was the incorporation of variables related to the sedimentological properties, such as sediment dry bulk density, porosity, grain size parameters, and mineral surface area. These additional variables are key for understanding the underlying reasons affecting the distribution of OC in continental margins. In general, OC is preferentially adsorbed to finer grained sediments with higher mineral-specific surface area (Keil et al., 1998; Mayer, 1994) (Fig. 4). Moreover, its mineral binding can also serve as a protective matrix that prevents the degradation of OC (Hedges and Keil, 1995; Hemingway et al., 2019). Hence, the hydrodynamic sorting of mineral particles due to differences in grain size affects the transport of OC, while regulating its ageing and degradation (Ausín et al., 2021; Bao et al., 2019).


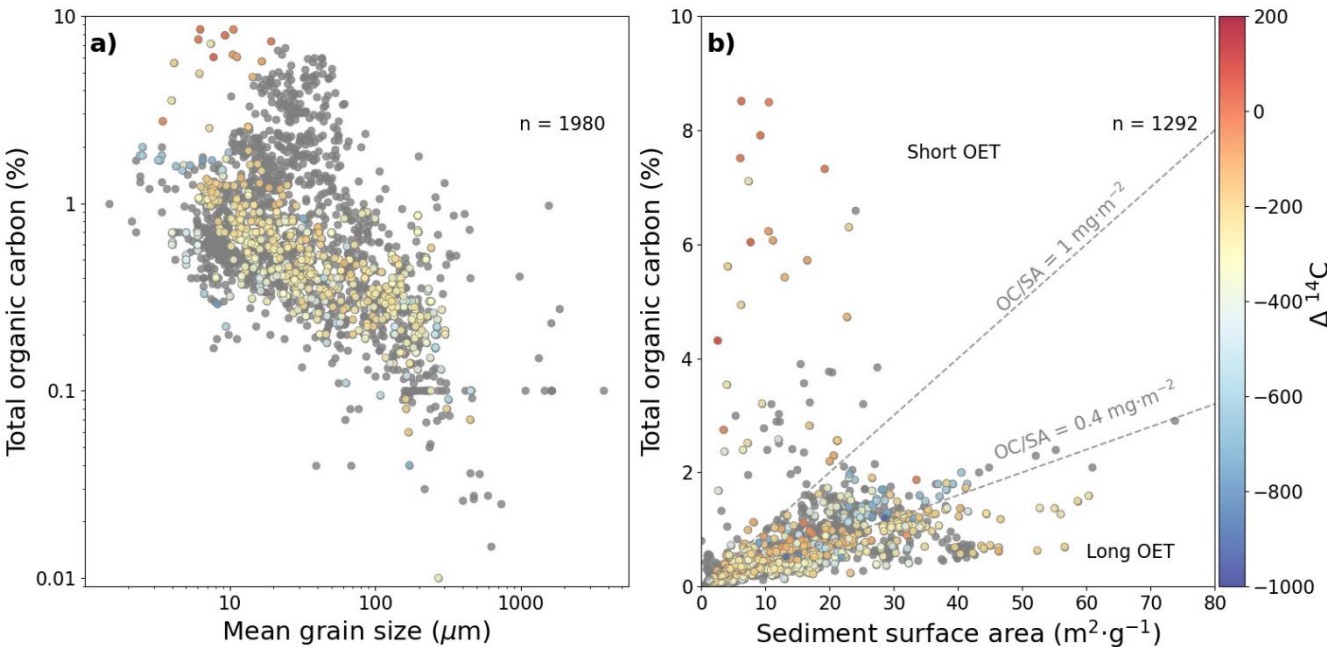


**Figure 4. Relationship between OC content and (a) sediment grain size fraction and (b) sediment surface area in marine sediments. The colour bar shows the ranges of $\Delta^{14}C$, if available (grey symbols indicate samples lacking concomitant $^{14}C$ data). Lines in (b) show the specific ranges of surface loading of different sedimentary environments (>1 mg·m-2, 0.4-1 mg·m⁻², <0.4 mg·m-2) based on the relative oxygen exposure time (OET), as explained by Mayer (1994).**

During the data compilation and harmonization, we noted that different labs use contrasting definitions for "clay fraction". While some laboratories define clay as particles that are smaller than 2 µm (e.g., Bruni et al., 2022; Schwab et al., 2021), others define it as the sediment fraction that is smaller than 4 µm (e.g., Hastings et al., 2012; Khan et al., 2020). To avoid confusion, two variables were created for the clay fraction, defining which threshold was used to define it (< 4 µm, < 2 µm). Moreover, during the data ingestion, grain size fractions are harmonized, calculating missing grain size fractions whenever possible to

enrich the dataset (see section 2.2.3 for more details).

Since different grain size fractions and density fractions present distinct sediment transport properties and may protect OC differently, several studies also analyse OC and its geochemical composition in different grain size or density fractions, which can be efficiently stored in MOSAIC v.2.0 by specifying the fraction analysed (column "material_analyzed" in the sample metadata table). For instance, the data stored in MOSAIC v.2.0 shows the dual effect of organo-mineral associations, where

the easily resuspended coarser silt fraction (20-63 µm) undergoes greater degradation and ageing of OC, while mineral surfaces in finer size fractions promote the protection of OC associated to these fractions (Fig. 5), a global process that occurs in all continental margins with different intensities depending on their depositional environments (Ausín et al., 2021; Bao et al., 2016, 2019; Bruni et al., 2022; Coppola et al., 2007).

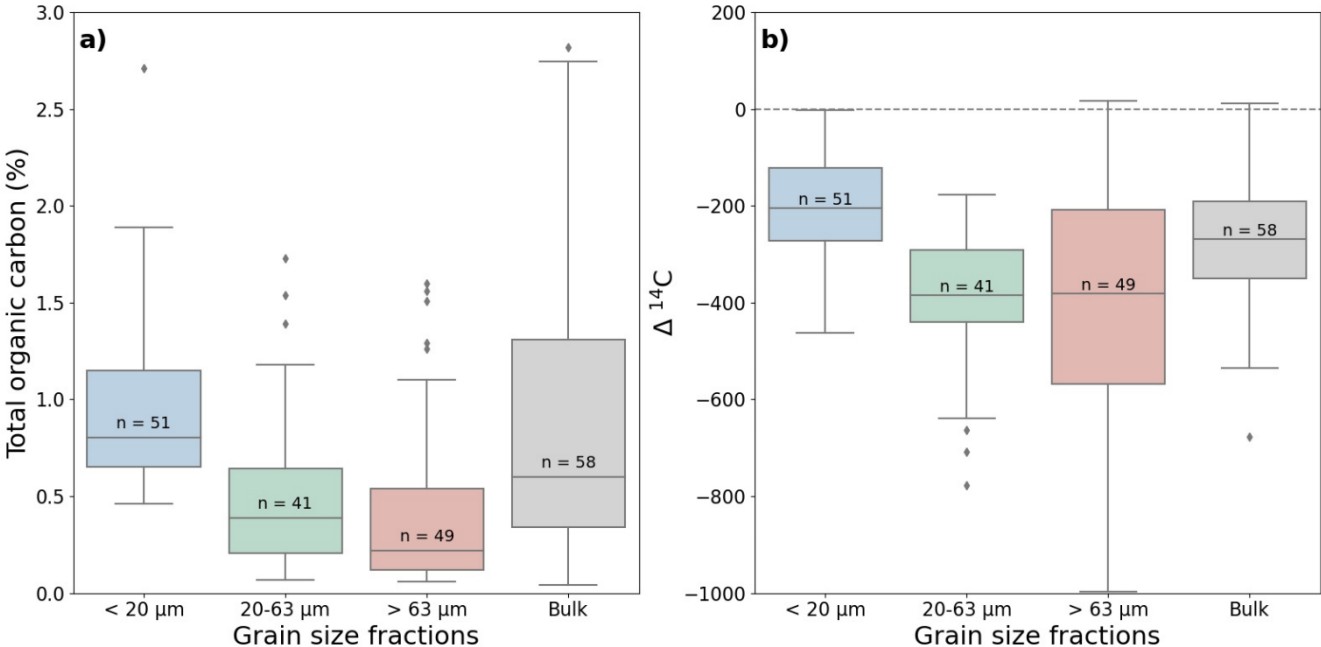

**Figure 5. Box plot of OC content (a) and Δ14C (b) in different grain size fractions, ranging from sand (> 63 µm), coarse silts (20-63 µm), and fine silt and clay (< 20 µm). The number of independent samples measured for each grain size fraction are annotated in each boxplot.**

### 3.2 Specific biomarkers

MOSAIC v.2.0 was also expanded to incorporate several groups of widely used biomarker compounds to better constrain the origin and degree of degradation of organic matter in continental margins worldwide. Although the variety of biomarkers measured in marine sediments is vast, we have focused on those that derive into lignin-derived phenols, long-chain alkanoic (fatty) acids, alkanes and alcohols, and alkenones given their wealth of existing data. Numerous prior contributions provide a full description of the origin and distribution of these biomarkers (e.g., Bianchi et al., 2018; Blair and Aller, 2012; Diefendorf and Freimuth, 2017; Sachse et al., 2012; Thevenot et al., 2010).

*Lignin phenols.* Lignin is a structural molecule that is almost exclusively found in the tissue of vascular (land) plants, and is thus used as a tracer of terrestrial biogenic organic matter in marine sediments (Bröder et al., 2016; Goñi and Hedges, 1990; Gordon and Goñi, 2003; Hedges and Mann, 1979; Prahl et al., 1994; Tesi et al., 2007). Lignin-derived phenols produced from oxidative alkaline hydrolysis of samples (Hedges and Ertel, 1982) can be separated into three main compound classes based on their molecular structure and origin: vanillyl (or guaiacyl) phenols (VP; angiosperms and gymnosperms), syringyl phenols (SP; gymnosperms), and cinnamyl phenols (CP; non-woody grasses) (Hedges and Mann, 1979). In addition to lignin phenols, cutin acids are also important tracers of terrestrial OC since they are only present in non-woody grasses and leaves (Goñi and Hedges, 1990). Given their distinct origin, ratios of the different phenols (SP/VP, CP/VP) can help elucidate the origin of plant sources (Goñi et al., 1998, 2000), although it can also be affected by hydrodynamic sorting of particles enriched in SP and CP



relative to VP (Bianchi et al., 2002; Pasqual et al., 2013). Similarly, the different proportions of acid and aldehydes within the vanillyl and syringyl phenolic groups can also provide an indication of the degree of degradation of terrestrial organic matter, given the higher reactivity of aldehydes with respect to acids (Gordon and Goñi, 2004; Tesi et al., 2012).

*Long-chain n-alkyl lipids.* Fatty acids, alkanes, and alcohols are naturally present in both marine and terrestrial organic matter, but with variable carbon chain lengths. In general, fatty acids, alkanes, and alcohols produced by marine organisms tend to be comprised of less than 24 carbon atoms, and are often referred to as low molecular weight (LMW) lipids. In contrast, terrestrial vascular plant leaf waxes are typically characterized by longer chain lengths ($\geq$ 24 carbon atoms), and are hence referred to as high molecular weight (HMW) (Eglinton and Hamilton, 1967). These compounds have been used to elucidate the sources of organic matter in marine sediments, as well as determine the contribution of organic matter from anthropogenic activities (Bai et al., 2021; Feng et al., 2013; French et al., 2018; Mead and Goñi, 2006).

*Alkenones.* Long-chain (typically $> C_{35}$) unsaturated ketones (alkenones) are produced by a specific kind of marine phytoplankton, coccolithophores, and are well preserved in marine sediments. These compounds serve as useful proxies of marine primary productivity (Raja and Rosell-Melé, 2021) as well as for the reconstructions of past sea-surface temperatures (Eglinton et al., 2001; Marlowe et al., 1984; Tierney and Tingley, 2018; Volkman et al., 1980).

Abundances of these biomarkers are often reported based on bulk sediment concentration (as e.g. µg per g dry weight), or normalized by the OC content of the sample (as e.g. µg per g OC), which is specified in the metadata stored in MOSAIC v.2.0 as the material analyzed (bulk sediment, OC, sediment fractions, etc.). Hence, concentrations of the biomarkers can be provided in either format. This data storage architecture also allows an efficient storage of data from compound-specific stable isotope analysis (CSIA), or compound-specific radiocarbon analysis (CSRA), providing a detailed overview of the sources and pathways of organic matter deposited in marine sediments (Feng et al., 2013; French et al., 2018; Gibbs et al., 2020; Gustafsson et al., 2011; Hahn et al., 2017; Huang et al., 2000; Kusch et al., 2010; Tao et al., 2016; Wakeham and McNichol, 2014; Yu et al., 2022).

For instance, since individual biomarkers have distinct sources, transit times, and reactivities, CSRA is useful to determine the diagenetic state of organic matter and depositional processes of specific biomarkers, which would be masked when analysing only the bulk OC (Eglinton et al., 1997; Kusch et al., 2021). The initial compilation of compound specific [14]C data for lignin phenols, fatty acids, alkanes, alcohols, and alkenones stored in MOSAIC v.2.0. underlines the contrasting radiocarbon age offsets due to the different origins and reactivities (Fig. 6). For instance, while bulk analyses show $\Delta^{14}C$ values that range between ~-600 and ~50 ‰, $\Delta^{14}C$ values of terrestrial biomarkers are significantly lower than for marine biomarkers, indicating older radiocarbon ages of terrestrial organic matter depositing in marine sediments. This trend generally occurs since terrestrial OC has a longer transit time from its production in terrestrial environments until its deposition in marine sediments in comparison to marine OC, whose biomarkers tend to retain the [14]C signal from the surface ocean which is closely-coupled with the atmospheric signal. In the case of terrestrial biomarkers, there is a progressive [14]C depletion among HMW (plant wax) compound classes in the order *n*-alcohols > *n*-fatty acids > *n*-alkanes which is attributed to the contrasting reactivity of these compounds (see review by Kusch et al. (2021)). Similarly, different marine biomarkers show different OC ageing due to their


different reactivities and transit times prior to deposition on the seafloor (Bröder et al., 2018; Feng et al., 2013; Hu et al., 2014; Mollenhauer and Eglinton, 2007; Tao et al., 2016; Wakeham and McNichol, 2014; Yu et al., 2022). These contrasting

radiocarbon ages in biomarkers help shed light on the origin and biogeochemical processes affecting the distribution of OC, which would be masked if only the bulk sediment was analysed.

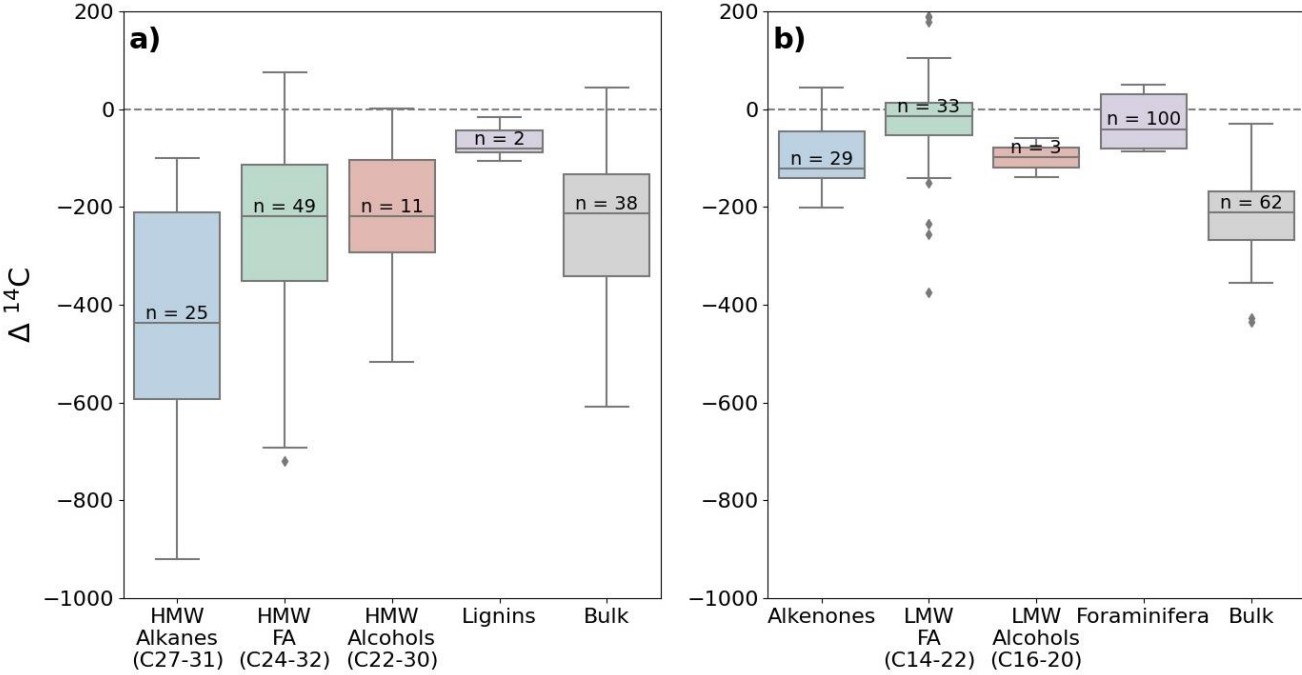

**Figure 6.** Box plot of compound-specific radiocarbon analyses (shown as $\Delta^{14}C$ values) for different terrestrial (a) and marine (b) biomarkers. Radiocarbon analyses in planktonic foraminifera are also provided to contextualize $\Delta^{14}C$ values of the other marine

biomarkers. The number of independent samples measured for each compound class are annotated in each boxplot. Note that the boxplots may encompass $\Delta^{14}C$ values of several individual compounds from the same sample. FA= Fatty acids.

### 3.3 Future expansions

MOSAIC v.2.0 considerably broadened the range of variables, including those that could account for the effect of hydrodynamic processes and mineral protection of OC (section 3.1), as well as specific biomarkers that could further refine

its sources (section 3.2). However, future versions of MOSAIC will expand the breadth of variables even further to improve our understanding of the processes affecting the fate of OC in marine sediments. For instance, the inclusion of additional variables such as clay mineralogy, concentration of major (e.g., Al) and trace (e.g., Nd) metals, as well as their isotopes, can provide additional insights into sediment provenance and transport pathways along continental margins (Blanchet, 2019; Fagel, 2007; Jeandel et al., 2007; Li et al., 2023; Liu et al., 2010; Schwab et al., 2021). Additional source-specific biomarkers such

as glycerol dialkyl glycerol tetraethers (GDGTs) (Damsté et al., 2002; Koga et al., 1993), long chain alkyl diols (de Bar et al., 2020), and sterols (Tao et al., 2022) could further define the origin of organic matter, while other biomarkers such as algal-derived pigments, biopolymeric fraction of carbon, amino acids and carbohydrates can determine its degree of reactivity





(Burdige and Martens, 1988; Dauwe and Middelburg, 1998; Pusceddu et al., 2009; Raja and Rosell-Melé, 2022). Additionally, new proxies are continuously being proposed that can further disentangle the source of organic matter (Lattaud et al., 2021),

and help refine the use of biomarker proxy calibration (Tierney and Tingley, 2014, 2018) which can be affected by sediment redistribution and degradation processes (Ausín et al., 2022; Lattaud et al., 2022). Future efforts will be directed into including these variables into MOSAIC to gain a holistic understanding of the fate of organic matter in marine sediments.

## 4 Spatiotemporal coverage of MOSAIC v.2.0

Although the main spatial focus of MOSAIC is the continental margin to understand the processes that affect carbon cycling

and burial in these heterogenous and complex areas, its spatial coverage extends to marine sediments on a global scale.

The number of individual locations for which data is stored in MOSAIC v.2.0 is now 17147, compared to ~4000 in the first iteration, filling in data gaps such as in the Chukchi Sea, Bering Strait, the Gulf of Mexico, Greenland and Norwegian seas, N Atlantic, SW Atlantic, Mediterranean, Persian Gulf, Bay of Bengal, Gulf of Thailand, South China Sea, Australasia, among others (Fig. 7a). Despite more than quadrupling the number of sampling locations stored in this new iteration, there remain

substantial gaps in certain areas, such as the continental margins of eastern Africa and Madagascar (western Indian Ocean), and Mesoamerica (Fig. 7a). This lack of independent and identical distribution of data (i.e., an even spread of datapoints), of marine sediments on a global scale can skew spatial analyses to perform well in oversampled areas, yet poorly in underrepresented areas (Meyer and Pebesma, 2022). While data exploration and compilation remain far from complete and on-going, we emphasize that more effort should be made to sample unrepresented sites if we want to produce reliable maps of

the global distribution of OC and other geochemical properties (Fig. 7a).

In order to study short-term temporal variations in the geochemical composition of OC in surficial marine sediments collected from different field campaigns, we have included the sampling date in MOSAIC v.2.0. Out of the 17147 sediment cores stored in the database, 74 % provide information of its sampling year, spanning from the 1950s until 2020 (Fig. 7b). According to the data available in MOSAIC v.2.0, the number of sediment cores collected during the last decades increased drastically, from

~500 sediment cores during the 1950s, to >9000 since the 1990s. However, the greater availability of published data and its digitalization over the last decades likely generates an inaccurate impression that more sampling has occurred over the last decades, and highlights the need to continue to digitize hard-copy data collected prior to the 1990s. Nevertheless, the large number of sediment cores collected over these decades underline the need for greater efforts to build and maintain curated databases and ensure harmonized and machine-readable data, which represent a significant return in investment of these

numerous oceanographic cruises (Lee et al., 2023). Hence, this is an invaluable dataset that will not only allow us to understand the spatial distribution of OC and its composition, but also assess how they have been changing over the last decades.

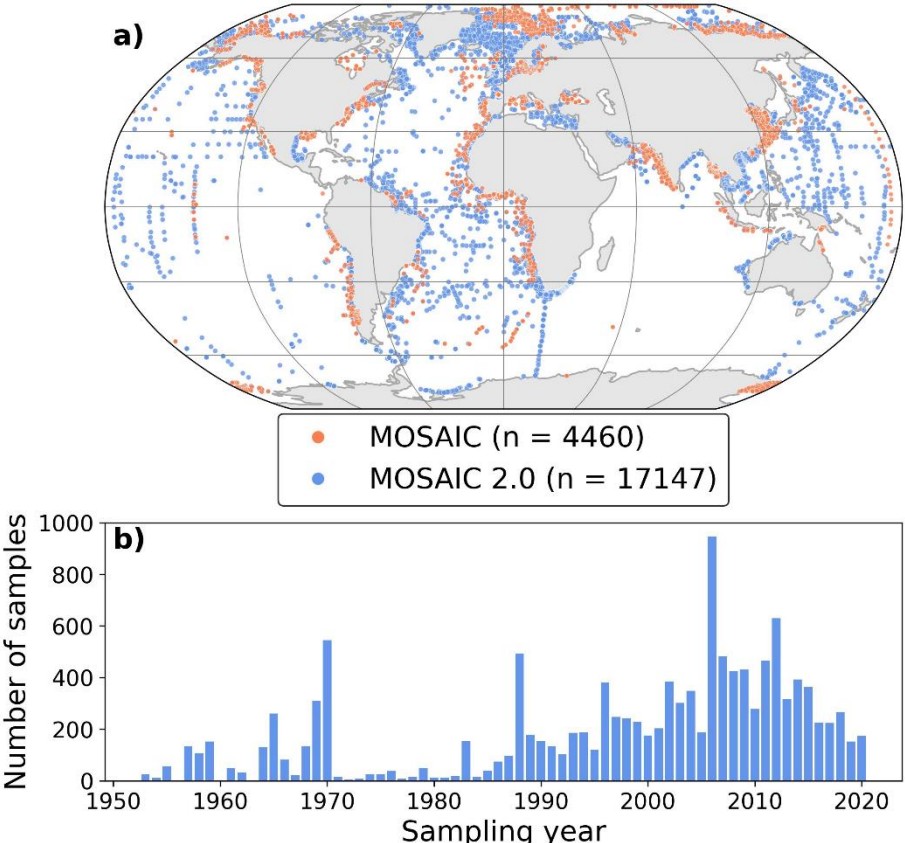

**Figure 7. a) Spatial distribution of sampling locations stored in the first iteration of MOSAIC (red) and additional data in MOSAIC v.2.0 (blue). Note the increase in spatial coverage for MOSAIC v.2.0. b) Temporal distribution of the datapoints in MOSAIC v.2.0. The previous iteration of MOSAIC did not store information of the sampling year.**

Despite the widespread geographical distribution of sampling locations in MOSAIC v.2.0, the spatiotemporal coverage of the variables stored in the database is relatively limited given the specificity of their analyses and limited number of laboratories that can conduct each analysis. We are aware that the spatial extension of certain variables would substantially increase by performing a thorough systematic review of the available literature, and future versions of MOSAIC will be focused on this.

In addition, we propose to further expand the spatiotemporal coverage of these less represented parameters by recovering legacy samples and performing additional analyses, circumventing the high costs and logistics of organizing and executing oceanographic cruises. Finally, we are also aware that the data stored in MOSAIC originate from English-based scientific journals and reports, which bias the availability of data. Future efforts should be directed to retrieve what is likely to be a wealth of data residing in journals, reports and data repositories written in other languages.

Below, we outline the spatiotemporal coverage of the main subgroups of variables: bulk and isotopic compositions, sedimentological properties, and biomarkers.



## 4.1 Distribution of bulk and isotopic compositions

As expected, the majority of the sampling locations stored in MOSAIC v.2.0 have data of sedimentary OC content, covering nearly all continental margins (Fig. 8a). In contrast, sediment cores with data of $CaCO_3$ and total nitrogen (TN) contents, and

the isotopic composition of organic matter ($\delta^{15}N$, $\delta^{13}C$, and $\Delta^{14}C$) are less extensive. Despite the reduced spatial distribution of these variables, the number of locations has increased substantially since the last iteration of MOSAIC (van der Voort et al., 2021).

Nearly half of the sediment cores in MOSAIC v.2.0 have $CaCO_3$ content data, partly due to harmonization techniques that calculate variables when sufficient information is provided (see section 2.2.3 for more details). Sediment cores with TN and

$\delta^{13}C$ generally have a similar spatial distribution, with good spatial coverage in the Arctic, Asian, Arabian, American, west African, and Australian margins, but to a lesser extent than OC and $CaCO_3$, and with much fewer sampling points in the open ocean (Fig. 8c, d). Finally, sediment $\delta^{15}N$ data is less well represented, with a sparse distribution of datapoints in the western South American, North American, west African, Asian, and Australian margins, as well as in the Mediterranean Sea, (Fig. 8f). Given the high costs and limited number of laboratories that can analyse $^{14}C$, the spatial coverage of radiocarbon data is the

least comprehensive (Fig. 8f). Given the radiocarbon-centric nature of this database, substantial efforts have been invested in compiling published radiocarbon analyses performed in marine sediments, and the spatial distribution of this variable has increased the most in this new iteration of the database, more than doubling from ~500 datapoints to ~1500, filling in regions surrounding the African, American, Asian and Arctic continental margins that were not represented in the previous iteration of MOSAIC.

The lower representation of isotopic analyses in sediment cores stored in MOSAIC highlights that more efforts are needed to recover legacy sediment cores from unrepresented margins and conduct additional analyses in these samples. Analysing TN, $\delta^{15}N$, $\delta^{13}C$, and $\Delta^{14}C$ in these legacy samples would help constrain the origin and age of organic matter, shedding light on the biogeochemical processes occurring along continental margins.





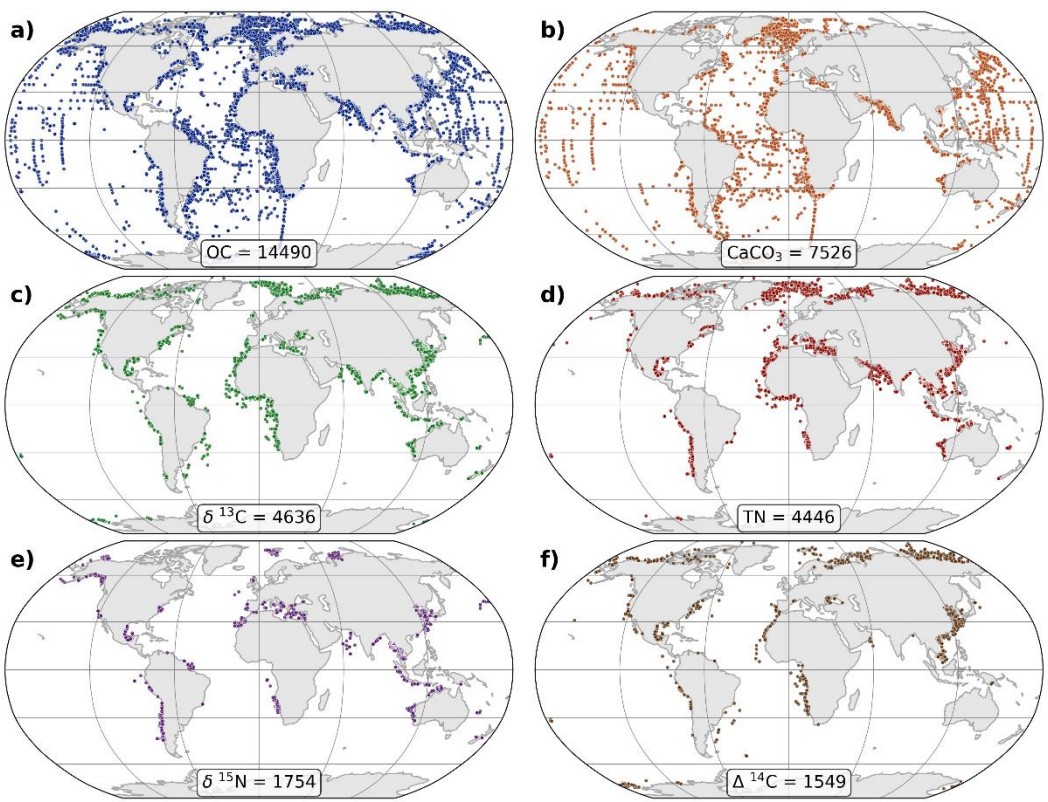

**Figure 8. Spatial distribution of sampling locations with surface and/or downcore data of (a) OC, (b) CaCO₃, (c) δ¹³C of OC, (d) TN, (e) δ¹⁵N in acidified or non-acidified sediment, and (f) Δ¹⁴C of OC.**

### 4.2 Distribution of sedimentological properties

As mentioned in section 3, the previous iteration of MOSAIC did not hold any data regarding the sedimentological properties of samples. Despite efforts to compile sedimentological data, the spatial coverage of sedimentological parameters in MOSAIC v.2.0 remains limited in comparison to OC (Figs. 8-9). Further effort will focus on compiling and harmonizing these data from continental margins, since there is clear spatial bias in the data available.

For instance, dry bulk density is a crucial parameter to calculate the OC stock in marine sediment, but this variable is seldomly reported (Fig. 9a), forcing researchers to infer it from other variables (Atwood et al., 2020; Diesing et al., 2017, 2021; Smeaton et al., 2021). We therefore urge researchers to measure and report dry bulk density in order to properly calculate carbon stocks.

Regarding variables related to sediment grain size, although the majority of studies measure grain size distributions, there is no consensus of how such data should be reported. While some studies only report mean (or median) grain size values (Fig. 9b), others report the relative proportion of different grain size classes (e.g., sand, silt, clay). However, as mentioned earlier, there are contrasting definitions for clay sizes (Fig. 9c-d). These different reporting strategies for grain size data lead to a broad spatial coverage of grain size analyses which cannot easily be harmonized. Similarly, some studies report mineral surface area





(Fig. 9e), which is linked to both grain size as well as mineralogy, so a harmonization between these variables should be explored to further expand the richness of this database.

Finally, as an effort to understand the effect of hydrodynamic sorting in the distribution of OC in marine sediments, MOSAIC v.2.0 also includes the possibility of adding geochemical analyses performed on specific grain size and density fractions. However, studies assessing this are currently very limited and only cover the East China Sea (Bao et al., 2018, 2019; Wang et

al., 2015), North American margin (Ausín et al., 2021; Coppola et al., 2007; Wakeham et al., 2009), and the Namibian margin (Ausín et al., 2021; Bruni et al., 2022) with a few additional sampling locations on the Peruvian, Iberian, north African, and New Zealand margins (Ausín et al., 2021; Bergamaschi et al., 1997; Cui et al., 2016) (Fig. 9f). In order to understand the global effect of grain size sorting and mineral protection, there is a need to expand these analyses in continental margins with different environmental conditions.

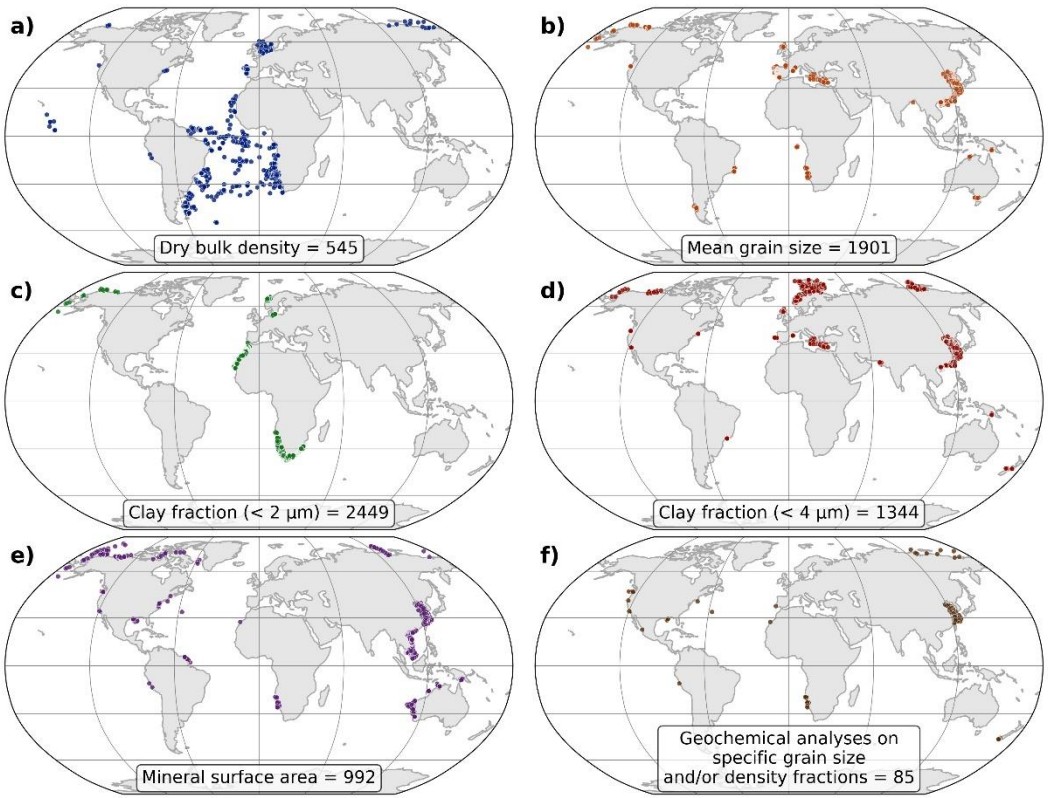


**Figure 9. Spatial distribution of sampling locations with surface and/or downcore data of (a) dry bulk density, (b) mean grain size, (c) clay fraction (< 2 µm), (d) clay fraction (< 4 µm), (e) sediment surface area, and (f) geochemical analyses in specific grain size and density fractions.**

### 4.3 Distribution of biomarkers

As with variables related to sedimentological properties, the previous iteration of MOSAIC did not hold any biomarker data. Despite on-going efforts to compile data of biomarkers, its spatial coverage in MOSAIC v.2.0 remains very limited, largely





due to the time-intensive nature of compiling this complex multivariate data for ingestion into the database. Instead of an exhaustive literature search for biomarker data, we highlight locations where data of these biomarkers are currently available in MOSAIC v.2.0. We recognize that extensive studies on many continental margins have been undertaken, and efforts are on-going to compile and harmonize this data into future versions of MOSAIC. Nevertheless, the goal in MOSAIC v2.0 is to develop a framework for ingestion and organization of biomarker data such that the community can contribute to, and provide feedback on, its subsequent expansion.

Of the biomarkers included in MOSAIC v.2.0, lignin phenols have the greatest spatial distribution, with data available from the North American margin, offshore the Amazon basin, the Arctic margins, Baltic Sea, East Asian margin and in some areas of the European, Arabian, Indonesian and New Zealand seas (Fig. 10a). Despite the greatest coverage of lignin phenols, these data almost exclusively derive from the Northern Hemisphere, limiting a proper understanding of controls on a global scale of this group of biomarker compounds, and efforts are on-going to compile this data together. Alkanes have been extensively analysed along the Alaskan, East Asian and equatorial west African margins, as well as in the Gulf of Mexico and South Georgia islands, but the database does not yet include any data from other continental margins (Fig. 10b). Included alkenone data are currently distributed along the eastern South American margin, in the Alaskan margin, and in the East Asian margin. Some isolated datapoints have been analysed as well in the north western African margin, Japanese margin and in the Pacific (Fig. 10c). Alcohols and fatty acids are the least represented biomarkers in MOSAIC v.2.0. The database only has data for fatty acids in the East Asian margin and in certain sites in the North American and Arctic margins and in South Georgia islands (Fig. 10d). In contrast, the database has data of alcohols only in the East Asian margin (Fig. 10e). Finally, data on compound-specific stable isotope and compound-specific radiocarbon analyses in MOSAIC v.2.0 is thus far only available along the North American, Iberian, west African, Arabian, Indian, East Asian and Arctic margins as well as in the Baltic and Black seas (Fig. 10f).

The limited data coverage of the targeted groups of biomarkers that have been ingested thus far in MOSAIC v.2.0 highlight the need to compile and harmonize published biomarker data from all continental margins, as well as to analyse available samples from under-sampled or under-reported regions in order to add this valuable source of information on the origin of OC in continental margins and the biogeochemical processes that occur along them. As mentioned earlier, future versions of MOSAIC will be focused to include more biomarkers (see section 3.3), and link existing databases of marine biomarker proxies (de Bar et al., 2020; Tierney and Tingley, 2015, 2018), while expanding their spatiotemporal coverage in MOSAIC.





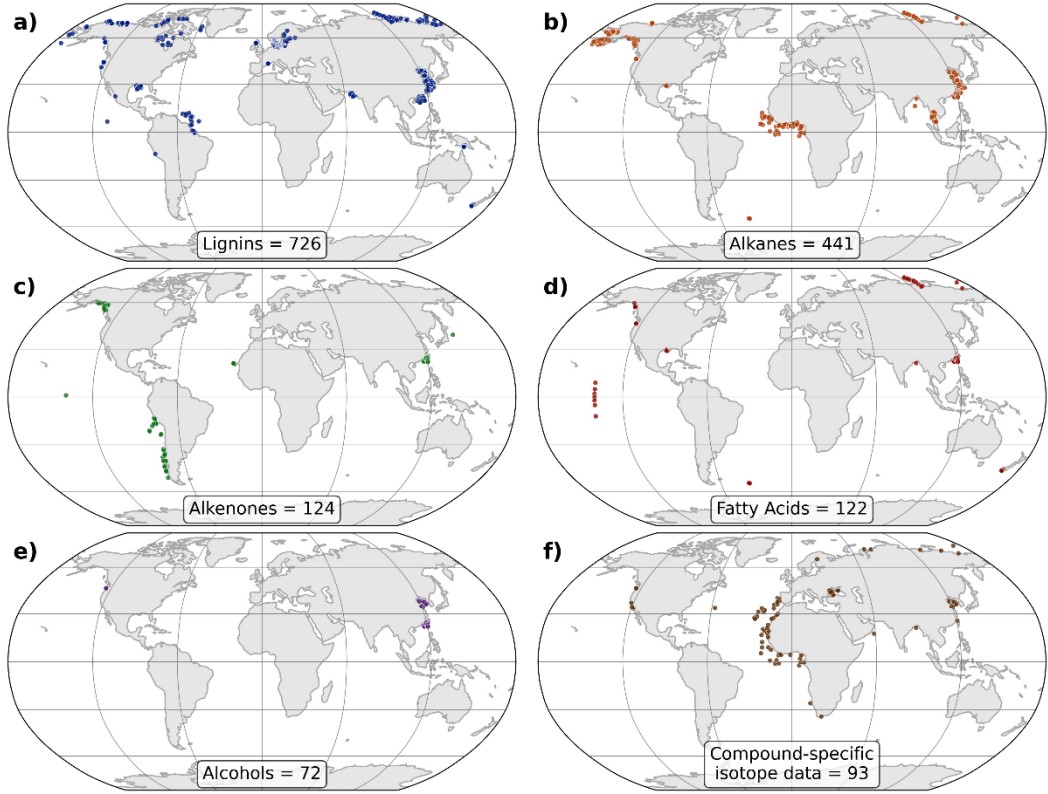

**Figure 10. Spatial distribution of sampling locations with surface and/or downcore data of (a) lignins, (b) alkanes, (c) alkenones, (d) fatty acids, (e) alcohols, and (f) compound-specific stable isotopic or radiocarbon analyses in any of the above-mentioned compounds.**

## 5 Data accessibility and version control

As in the previous MOSAIC version (van der Voort et al., 2021), the SQL data can be found in the supplementary information and through the website (doi:10.5168/mosaic019.1; mosaic.ethz.ch), where users can interactively visualize the data in plots as well as download a subset of the database. Since its publication, the website presents further functionalities that will promote the user's interaction with the database: 1) downloading the whole spatial extension of the database in a spreadsheet, 2) downloading the input template, and 3) uploading the template with new data to contribute to the database. Given the complexity of the new data stored in MOSAIC v.2.0, only selected analyses are available on the website, but users can download the full SQL database from the supplementary information.

## 6 Conclusions

We present here a new version of MOSAIC, v.2.0, which compared to the initial version, has expanded its spatiotemporal coverage by ~400 % and incorporates additional variables (e.g., sedimentological properties and biomarkers) to facilitate a

more comprehensive understanding of the processes that affect the distribution and degradation of organic carbon from different sources in marine sediments. In addition, this new database includes richer metadata that maximize the comparability of data, complying with FAIR data principles. While this new version of the database includes data from more than 17000 individual sediment cores, further efforts are needed to compile and harmonize data from thus far unsampled areas to better understand the distribution of OC in marine sediments on a global scale and apply novel machine learning techniques to identify different depositional environments and the factors that affect the distribution of organic carbon in marine sediments. Since this database is a collaborative effort, we urge the scientific community to continue to contribute to this growing database, which will further enhance its value and the research outputs it can provide.

**Author contribution**

SP, TSvdV and TE discussed the expansion of the MOSAIC database. SP, KN and AW collected data and contributed to the discussion of the data ingestion. SP restructured the database and perfectioned the ingestion pipeline through automated Python scripts. SP and TSvdV adapted the RShiny website for this exansion. HG, TB, LB and TE contributed to the discussion of appropriate metadata stored for each variable. SP wrote the manuscript with the help of all co-authors.

**Competing interests**

The contact author has declared that none of the authors has any competing interests.

**Acknowledgements**

The production and implementation of this database was funded by the Swiss National Science Foundation funded project Climate and Anthropogenic PerturbationS of Land-Ocean Carbon TracKs (CAPS-LOCK3; SNF200020_184865/1). We would like to thank all the researchers who provided their data in tabular format: Meixun Zhao, Hu Limin, Yu Fengling, Ge Chendong, Kostas Kariakoulakis, Henko de Stigter, Rut Pedrosa-Pàmies, Miguel Goñi, Chun Zhu, and Markus Diesing. Their input contributed greatly to increase the spatiotemporal content of this database. Special thanks go to Matthias Brakebusch, whose discussions have provided great insight into the database's structure.

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
