# Peer review of "The Modern Ocean Sediment Archive and Inventory of Carbon (MOSAIC): version 2.0"

_Earth System Science Data, 2023_

## Referee Comment (RC2)

[referee-annotated manuscript omitted]

---

## Author Response (AR1)

**Response to Reviewer 1**

We would like to thank the reviewer for their praise in this new version of the MOSAIC database, which has grown substantially since its first iteration and has improved its metadata structure to align to the latest reporting strategies.

As the reviewer well notes, we highlight which variables are missing, such as the insufficient spatial coverage of dry bulk density. In the new revised version, the revised manuscript expands the importance of having a greater spatiotemporal coverage of certain under-represented variables (e.g. $\Delta^{14}C$ and mineral surface area).

In addition, we provide a list of parameters that we urge researchers to report in their studies that will improve the quality of data stored in the database, such as sampling method, sampling dates, and analytical methods. The amended manuscript now reads:

*"We urge researchers to provide sufficient metadata in their studies (e.g., sampling technique, sampling dates, and details of analytical methods) that enhances the quality and utility of this database."*

**Response to Reviewer 2**

We would like to thank Dr. Helen Bostock for her thorough review of the manuscript and her praise in this updated version of the database, which has increased substantially since its first iteration and has incorporated methods to ensure minimal duplication and improve data comparability. We have addressed all the comments and suggestions Dr. Bostock has kindly given us, which are incorporated in the amended manuscript and explained in the sections below:

We have included a new supplementary figure (Fig. S4) with a screenshot of the database's dashboard, as well as the online dataset, which is shown below:

[Figure]

Dr. Bostock is correct that we should provide more emphasis on the big questions that the database can be used for. Highlighting the different uses of the database may encourage researchers to provide their data, making this a more collaborative database. This has been included in the amended manuscript as the following sentence:

*"While MOSAIC can be used to model global distribution of OC content (Atwood et al., 2020) and identify vulnerable sites of OC disturbance (Clare et al., 2023), it can also provide a global context of the geochemical characteristics of a specific study area (Bruni et al., 2022), and even locate suitable sites and samples that can answer specific research questions (N. Golombek, pers. comm.., 2023)."*

As Dr. Bostock points out, we mainly focus on how the database can be used to assess spatial variations in the geochemical properties of marine sediments, without providing much emphasis on the temporal variations, even though we refer to the spatiotemporal nature of the database. We have added some brief examples of how this database can be used to assess temporal variations in the geochemical properties of marine sediments in the amended version of the manuscript, which now reads:

*"However, these sediment cores were collected in different years. Current global modelling approaches combine data collected in different decades in order to increase spatial coverage, but ignore evidence that OC in surficial sediment has been changing in recent years due to direct (e.g., demersal fisheries or mining; Keil, 2017; Paradis et al., 2019,2021b; Clare et al., 2023) or indirect (e.g., land use changes and climate change; Bröder et al., 2021) anthropogenic impacts, which impact virtually the entire marine environment (Halpern et al., 2008). In order to study temporal variations in the geochemical composition of OC in surficial marine sediments over the last century, we have included the sampling date in MOSAIC v.2.0."*

Regarding the spatial scope of the database, we have included the following sentences in the amended version of the manuscript:

*"This includes samples collected in estuaries, inner and outer shelves, slopes, abyssal plains, and sediment from the open ocean, but currently excludes intertidal areas and complex blue carbon ecosystems such as mangroves and salt marshes."*

Since Dr. Bostock considers that the inclusion of biomarker data in MOSAIC is unclear, we provide more detail on how this data is processed and stored in the database:

*"These biomarkers are stored in separate tables based on their general compound classes. Since characteristics of total long-chain n-alkyl lipids (e.g., fatty acids, alkanes, and alcohols) depend on the specific carbon chain lengths measured, we specify the measured homologues in the method details to improve the comparability of data. In addition, MOSAIC v.2.0 stores the individual concentrations of specific homologues (e.g., $C_{16}$ fatty acids, $C_{18}$ fatty acids, $C_{29}$ alkanes) to allow researchers to calculate total concentrations within specific carbon chain lengths (e.g., HMW or LMW)."*

Finally, we have incorporated the annotations and comments Helen Bostock has provided in the manuscript, some of which are highlighted below:

- We have added over 4000 geopoints from datasets Dr. Bostock has provided us, further expanding the spatiotemporal breadth of the database.
- We have specified that, even if the database is carbon-centric, datasets are not required to have carbon measurements in them in order to be included in MOSAIC. The database incorporates any variable that helps understand the distribution of carbon in marine sediments on a global scale, and the new data ingestion algorithms help link datasets that

were published in different studies. Hence, the inclusion of datasets without any carbon measurements may later be linked to datasets with carbon measurements.

- We have also specified that the new "material analyzed" column allows the inclusion of $\Delta^{14}C$ measurements in both organic carbon or inorganic carbon fractions (e.g. foraminifera).
- We have also specified throughout the manuscript that the database includes both surficial and downcore datasets.